# Molecular Cloning, Characterization, and Expression Regulation of Acyl-CoA Synthetase 6 Gene and Promoter in Common Carp *Cyprinus carpio*

**DOI:** 10.3390/ijms21134736

**Published:** 2020-07-03

**Authors:** Dizhi Xie, Zijie He, Yewei Dong, Zhiyuan Gong, Guoxing Nie, Yuanyou Li

**Affiliations:** 1College of Marine Sciences of South China Agricultural University & Guangdong Laboratory for Lingnan Modern Agriculture, Guangzhou 510642, China; xiedizhi@scau.edu.cn (D.X.); jie3719@126.com (Z.H.);sunshinewonder@163.com (Y.D.); 2Laboratory of Aquatic Animal Nutrition and Diet, College of Fisheries, Henan Normal University, Xinxiang 453007, China; 3Department of Biological Sciences, National University of Singapore, Singapore 115473, Singapore; dbsgzy@nus.edu.sg

**Keywords:** acyl-CoA synthetase 6, transcriptional regulation, DHA deposition, *Cyprinus carpio*

## Abstract

Omega-3 long chain polyunsaturated fatty acids (n-3 LC-PUFA), particularly docosahexaenoic acids (22:6n-3, DHA), have positive effects on multiple biologic and pathologic processes. Fish are the major dietary source of n-3 LC-PUFA for humans. Growing evidence supports acyl-coenzyme A (acyl-CoA) synthetase 6 (*acsl6*) being involved in cellular DHA uptake and lipogenesis in mammals, while its molecular function and regulatory mechanism remain unknown in fish. The present study focused on investigating the molecular characterization and transcription regulation of the *acsl6* gene in the freshwater teleost common carp (*Cyprinus carpio*). First, the full length of *acsl6* cDNA contained a coding region of 2148 bp for 715 amino acids, which possessed all characteristic features of the acyl-CoA synthetase (ACSL) family. Its mRNA expression was the highest in the brain, followed by in the heart, liver, kidney, muscle, and eyes, but little expression was detected in the ovary and gills. Additionally, a candidate *acsl6* promoter region of 2058 bp was cloned, and the sequence from −758 bp to −198 bp was determined as core a promoter by equal progressive deletion and electrophoretic mobility shift assay. The binding sites for important transcription factors (TFs), including stimulatory protein 1 (SP1), CCAAT enhancer-binding protein (C/EBPα), sterol-regulatory element binding protein 1c (SREBP1c), peroxisome proliferator activated receptor α (PPARα), and PPARγ were identified in the core promoter by site-directed mutation and functional assays. Furthermore, the intraperitoneal injection of PPARγ agonists (balaglitazone) increased the expression of *acsl6* mRNA, coupling with an increased proportion of DHA in the muscle, while opposite results were obtained in the injection of the SREBP1c antagonist (betulin). However, the expression of *acsl6* and DHA content in muscle were largely unchanged by PPARα agonist (fenofibrate) treatment. These results indicated that *acsl6* may play an important role for the muscular DHA uptake and deposition in common carp, and PPARγ and SREBP-1c are the potential TFs involved in the transcriptional regulation of *acsl6* gene. To our knowledge, this is the first report of the characterization of *acsl6* gene and its promoter in teleosts.

## 1. Introduction

Omega-3 long chain polyunsaturated fatty acids (n-3 LC-PUFAs), particularly eicosapentaenoic acid (EPA; 20:5n-3) and docosahexaenoic acids (22:6n-3, DHA), have positive effects on multiple biologic and pathologic processes including metabolic disorders, cardiovascular and neurological diseases, cancer, inflammation, and pain [1,2,3,4,5]. Biosynthesis of n-3 LC-PUFAs is from α-linolenic acid (ALA, 18:3n-3) substrates, but ALA is limited (less than 4%) in humans; hence, the minimum recommended intake of EPA and DHA is 0.2–0.45 g per day [1,6]. It is well known that the benefits of food fish are to provide n-3 LC-PUFAs in diets, while with wild fish resources declining, the major source of n-3 LC-PUFAs has turned to cultured fish [7]. To date, the available plant lipid sources (without LC-PUFAs but rich in C18 PUFAs) are widely applied in aquaculture to reduce the fish oil (rich in LC-PUFA) dependence and feeding cost, while these lipids reduce the muscular n-3 LC-PUFA level (especial for the DHA level) in tested fish species and greatly affect the nutritional quality of cultured fish [8]. To efficiently use plant-based alternatives and to maximize endogenous n-3 LC-PUFA biosynthesis, much attention has been focused on illuminating the regulation mechanisms of LC-PUFA biosynthesis in teleosts [9,10,11,12]. However, it is still unclear about the n-3 LC-PUFA deposition process and its regulatory mechanism in fish muscle.

It has been reported in mammals that LC-PUFAs are desorbed and diffused into cells through protein-mediated transport mechanisms, such as fatty acid translocase (FAT/CD36), fatty acid transport protein (FATPs) and plasma membrane fatty acid binding protein (FABPpm) [13,14,15]. Similar to the FATPs above, the long-chain acyl-coenzyme A (acyl-CoA) synthetase (ACSL) family, including Acsl1-6, has also been shown to accelerate fatty acid uptake through influencing the cellular fatty acid metabolism in peripheral tissues and cells [16,17]. ACSL enzymes play an important role in lipid metabolism, since these enzymes activate long chain fatty acid (LCFA, with 12–22 carbons) to form acyl-CoAs, which is subsequently used in almost all catabolic and anabolic downstream pathways of FA metabolism [16,17]. Additionally, five distinct ACSL genes differing in their substrate preferences have been identified in mammals. Recent studies prove that *acsl6* is required to specifically enrich DHA in the mouse brain and that the *acsl6* mutant (*acsl6*^−/−^) has a significant decrease in brain DHA level [18,19,20]. *Acsl6* is also involved in the selective enrichment of DHA-containing membrane phospholipids in differentiating spermatids and seminiferous tubules, thus supporting its role in normal spermatogenesis in mice [21,22]. Moreover, overexpression of *acsl6* stimulates the biosynthesis of phospholipid and triglycerides and decreases the mitochondrial oxidative capacity in mammalian skeletal muscle [23,24]. Thus, these mammal data could provide clues for studying the function of *acsl6* in the DHA deposition in fish muscle.

Common carp (*Cyprinus carpio*) is one of the most important cultured freshwater fish in the world, especially in China, and over 4 million tons are produced annually [25]. However, the muscular DHA content in farmed common carp was significantly lower than that in the wild fish [26,27]. To explore the molecular mechanism of DHA deposition in common carp muscle, the present study was focused on investigating the molecular characterization and the transcription regulation of the *acsl6* gene. First, the full-length cDNA of the *acsl6* gene and its candidate promoter were cloned. Second, key elements in the core promoter region were identified by the targeted mutation of potential binding sites of selected key transcriptional factors (TFs). Finally, agonist/antagonist injection was performed to identify the roles of key TFs. These results should provide novel insights into the regulatory mechanisms of muscular DHA deposition in fish.

## 2. Results

### 2.1. Sequence and Phylogenetic Analyses of Common Carp acsl6 cDNA

A cDNA (3643 bp) of common carp *acsl6* was obtained, including a 5′-untranslated region (UTR) (214 bp), a full-length coding region (2148 bp) encoding 715 amino acids (GenBank accession no. MT431660), and a 3′-UTR (1420 bp) (Appendix A). The deduced protein sequence possessed all typical active site residues of ACSL, including adenosine monophosphate monophosphate (AMP) binding sites, CoA binding site, and acyl-activating enzyme consensus motif, as well as a gate domain with D-x4-(F, Y)-LPLAH-x2-E and a linker motif with GD-x13-DR-x4 (Figure 1), which has been well conserved in common carp and other fish species possessing *acsl6*, includeing *Epinephelus lanceolatus*, *Nothobranchius furzeri*, *Oreochromis niloticus*, *Salmo solar*, *Carassius auratus*, and *Danio rerio* (Figure 1).

A neighbor-joining phylogenic tree was constructed based on the deduced *acsl6* amnio acid sequences from fish and other vertebrates. Phylogenetic analysis revealed that *acsl6* was divisible into four main clusters, including fish, birds, amphibians, and mammals (Figure 2). The common carp *acsl6* was grouped with orthologues from other teleost species and more distantly to other vertebrates, and it was clustered most closely to *C. auratus* (Figure 2).

### 2.2. Tissue Distribution of the Common Carp acsl6

RT-PCR and qPCR were used to analyze *acsl6* mRNA expression of in different common carp tissues (Figure 3A,B). The transcript of *acsl6* was detected in the brain, eyes, heart, liver, kidney, spleen, red muscle, white muscle, mesenteric fat, anterior intestine, middle intestine, and hind intestine but not in the gills and ovary (Figure 3A), with apparent higher expression levels in brain compared to the other tissues (Figure 3B).

### 2.3. The Structural Features of Common Carp acsl6 Gene Promoter

A 2058-bp candidate promoter of the *acsl6* gene was cloned in this study. Schematic diagram and sequence of the *acsl6* promoter are shown in Figure 4 and Appendix A, respectively. It includes 1790 bp of upstream nontranscribed sequence and 214 bp 5′-UTR. To determine the core promoter region of the *acsl6* gene, the full length of candidate promoter (SD1) and its 5′ serial truncations (SD2–SD4) were fused to a luciferase reporter vector pGL3.0 and tested for their ability to mediate transcription. Among the four promoter constructs, SD3 showed the maximal promoter activity. As SD4 had a significant decrease of promoter activity compared to SD3 (Figure 4), the region between SD4 to SD3 (−758 bp to −198 bp) was identified as the core regulatory region of the common carp *acsl6* promoter, and the sequence of this fragment was used for further functional analysis.

### 2.4. Demonstration of Binding of Muscle Nucleus Proteins to the Core Regulatory Region by EMSA

To further confirm whether the core regulatory regions of the *acsl6* promoter provides binding sites for potential TFs. Electrophoretic mobility shift assay (EMSA) was performed with muscular cytoplasmic and nuclear proteins. As shown in Figure 5, a gel shift band was observed only in lane 3 with nucleus proteins and 5′ biotin labeled probe, which indicated an interaction between nucleus proteins and the core regulatory region of *acsl6* (Figure 5). There was no gel shift band in other lanes, such as lane 1 without proteins; lane 2 with cytoplasmic proteins; or lane 4 with nucleus proteins, an unlabeled competitor probe, and a 5′ biotin labeled probe. These results indicated that some nucleus proteins in muscle could bind to the core promoter of common carp *acsl6*.

### 2.5. Identification of Cis-Acting Elements in the Core Regulatory Region of Promoter

Bioinformatics analysis of the core regulatory region of the common carp *acsl6* promoter indicated that there are eleven predicted TF binding sites including forkhead transcription factor 1 (FOXO1), stimulatory protein 1 (SP1), activated protein 1 (AP1), caudal type homeobox transcription factor 1 (CDX1), Yin Yang 1 (YY1), CCAAT enhancer binding protein (C/EBPα), TATA box binding protein (TBP), sterol-regulatory element binding protein 1c (SREBP1c), peroxisome proliferator activated receptor α (PPARα), and PPARγ (Figure 6). Also, these eleven predicted TF binding sites were mutated and transfected into Human embryonic kidney cell line (HEK 293T) cells for determining the effects on transcriptional activity. Compared with wild type SD3, mutation of TF binding sites for PPARγ, PPARα, and SREBP1c resulted in the most decreased transcriptional activity (*p* < 0.05), followed by SP1 and C/EBPα (Figure 7), which suggested that these TF binding sites were very important for maintaining the *acsl6* promoter activity.

### 2.6. Effects of TFs Agonists/Antagonist on TFs and acsl6 Expression and Fatty Acid Composition in Muscle

To further confirm the regulatory role of PPARγ, PPARα, and SREBP1 on *acsl6* expression, PPARα agonist fenofibrate, PPARγ agonist balaglitazone, or SREBP1c antagonist betulin were injected into the enterocoelia of juvenile common carp. The results of qPCR showed that the muscular *ppar*γ and *acsl6* mRNA levels significantly increased in balaglitazone treatment groups compared to the dimethyl sulfoxide (DMSO) group (control), and muscular *srebp1* and *acsl6* mRNA levels were significantly downregulated by antagonist betulin (*p* < 0.05) (Figure 8). Although the muscular level of the *pparα* mRNA expression level was upregulated by fenofibrate (*p* < 0.05), *acsl6* mRNA levels showed no significant difference between the fenofibrate group and the control group (*p*
*>* 0.05) (Figure 8).

Moreover, the results of fatty acid composition in muscle showed that there was a higher content of DHA in the balaglitazone group compared with that in the control group (*p* < 0.05) while the content of DHA in fenofibrate treatment was comparable with that in the control group (Table 1). Comparing with the control group, the muscular DHA, ARA, and 18:3n-6 were significantly decreased in the betulin group (*p* < 0.05) (Table 1).

## 3. Discussion

Several members of the ACSL family catalyze a wide range of fatty acids to form acyl-CoAs with some substrate preferences toward different FA metabolic fates [16]. To date, ACSL1, ACSL3, and ACSL4 have well been identified in numerous species, while *acsl6* has only been identified in rat, human, and mouse [16,28,29,30]. Several TFs, such as Sp1, LXR, SREBP, and PPARγ, contributing to the control and regulation of ACSL1, ACSL3, and ACSL4, have also been characterized in mammals [31,32,33,34,35]. In this study, the molecular characterization, tissue distribution, and transcriptional regulation of common carp *acsl6* were investigated and, here, provided evidences for the function of SREBP1 and PPARγ on *acsl6* expression and muscular DHA content in fish.

The present study represents the first characterization of *acsl6* cDNA in fish. The predicted common carp *acsl6* peptide possesses highly conserved residues of the ACSL family including the two well characterized domains, the gate and linker domains [36,37]. The gate domain (D-x4-(F, Y)-LPLAH-x2-E) contains an entry gate for the long chain fatty acid substrate [36,37]. The linker domain (GD-x13-DR-x4) of ACSL contains an ASP (D) residue, while a Gly (G) residue in that of short and medium ACS [38]. The overall tree branching pattern of *acsl6* indicated that the vertebrate *acsl6* sequence was divisible into four clades and that the common carp *acsl6* peptide was highly similar to other *acsl6* peptides from teleosts (Figure 2). This is in accordance with the recent finding that ACSL proteins have been highly conserved throughout evolution [17].

The gene expression pattern indicates its functional diversification in the tissue to some extent. In mammals, *acsl6* has a fairly restricted expression pattern but is highly expressed in brain and gonad [17,30] and has been suggested to be important and specific for DHA uptake in brain, differentiating spermatids, and seminiferous tubules [18,19,20,21,22]. Compared to mammals, a relatively widespread distribution pattern was observed in common carp *acsl6* mRNA, which showed the highest expression level in the brain and weaker expressions in the heart, liver, kidney, muscle, and eye but no expression in the ovary. This is similar to the tissue distribution obtained from zebrafish, in which the mRNA level of *acsl6* was greatest in the brain, followed by other tissues (eye, heart, testis, and kidney) and no expression in the ovary [17]. Between fish and mammals, the differences in tissue distribution of *acsl6* possibly reflected the functional importance of DHA for these tissues [2,22] and the high content of DHA in fish [7].

It is well known that the spatiotemporal expression pattern of gene expression is dependent on the binding of transcription factors to specific sequences in gene promoters [39]. In this study, a 2058-bp *acsl6* candidate promoter was also cloned from common carp and it included 1790 bp of upstream nontranscribed sequences and 214 bp of 5′-UTR without intron (Figure 4). Previously, mammal *acsl6* gene promoters had been reported in mice, which includes 2102 bp of upstream nontranscribed sequences and 443 bp of 5′-UTR with 334 bp introns [30]. The unidirectional deletion analysis showed that the upstream region of −2102 bp to +45 bp exerts a negative regulatory function to control mouse *acsl6* promoter activity [30] while the negative regulatory region of common carp *acsl6* promoter was located between −1789 bp and −758 bp. These above results suggested that the shorter negative regulatory region may contribute to the relatively widespread distribution pattern of the common carp *acsl6* gene. Based on progressive deletion, the core region of the common carp *acsl6* promoter was −758 to −198 (Figure 4). Additionally, the verification of EMSA also indicated that some muscular nucleus proteins could bind to the core promoter of common carp *acsl6* (Figure 5), while the core promoter of mouse *acsl6* was in the proximal promoter (+339 to +412) [30].

The mouse *acsl6* core promoter has been shown to contain binding sites for transcription factors AP2, SP1, C/EBPa, C/EBPb, CREB, and NF-1 [30]. Here, we showed that the common carp *acsl6* promoter shared the same binding sites for SP1 and C/EBPα. Site-directed mutagenesis revealed that cis-elements such as SREBP1c, PPARα, and PPARγ were essential in driving the activity of the common carp *acsl6* promoter (Figure 7), which was consistent with the activation of PPARα, PPARγ, and SREBP1c for mammalian Acsl1 and Acsl5 transcription, respectively [40,41]. Moreover, Motallebipour et al. [42] found that an SREBP-1 binding site was located in the first intron of the human *ASCL6* gene promoter. Those results indicated that the SRE and PPAR elements could also be essential in driving *acsl6* promoter activity in fish.

We also found that the PPARγ agonist (balaglitazone) induced *ppar*γ expression and simultaneously increased the expression of *acsl6* in muscle; in contrast, the SREBP1c antagonist (betulin) decreased the expression of *acsl6*, coinciding with the suppression of *srebp*1c expression (Figure 8). These results suggested a possible regulation of PPARγ and SREBP-1c on common carp *acsl6* expression. Although transcriptional control of the *acsl6* has not been well studied, transcriptional studies of another ACSL family indicated that the PPARγ agonist increased mammal *Acsl5* and *Acsl1* mRNA expression in the adipose and muscle tissue [43,44]. Hepatic *Acsl5* expression increased by SREBP-1c overexpression in transgenic mammals and hepatocytes [40,45]. These studies implied that PPARγ and SREBP-1c are common TFs for the ACSL family. The positive regulation of *acsl6* by TFs eventually embodied the changing products of the *acsl6* enzyme. Accordingly, in the present study, it was clearly shown that increased DHA concentrations of muscle were associated with the treatment of PPARγ agonist while the muscular DHA levels were decreased by SREBP1c antagonist treatment. In mammals, growing evidence supports a role for the *acsl6* in cellular DHA uptake [18,19,20,21,22] and muscular lipogenesis [23]. Collectively, these observations suggested that *acsl6* probably promotes muscular DHA uptake and deposition in common carp, which are subject to the regulation of PPARγ and SREBP-1c.

In conclusion, the present study has described the cloning and molecular characterization of the *acsl6* cDNA and promoter from common carp. Moreover, PPARγ and SREBP-1c were identified as the potential TFs for *acsl6* gene transcription in vertebrates for the first time. The results might contribute to the exploration of enhancing muscular n-3 LC-PUFA contents in farmed fish. Compared with wild fish, farmed fish decreased the chance of acquiring dietary DHA by formula feed with plant lipid sources. The dietary fatty acid compositions may also affect the *acsl6* expression, which further reduce the DHA content in farmed fish. Thus, further studies seem necessary to investigate the nutritional regulation of *acsl6* expression.

## 4. Materials and Methods

### 4.1. Ethics Statement

All fish procedures were performed following the National Institutes of Health guide for the care and use of laboratory animals (NIH Publications No. 8023, revised 1978) and approved by the Institutional Animal Care and Use Committee of the South China Agricultural University (SCAU-AEC-2010-0416, approved on 18 June 2018). All the fish surgeries were performed with 0.01% 2-phenoxyethanol (Sigma-Aldrich, St. Louis, MO, USA) anesthesia to minimize suffering.

### 4.2. Fish and Tissue Collection

Common carp juveniles (about 13.0 g) were purchased from a commercial aquafarm (Guangzhou, China). Before experiments, all fish were acclimated to the experimental conditions by feeding with a formula feed (32% crude protein, 8% crude lipid) for two weeks. The formulations and proximate and fatty acid compositions of diets are presented in Appendix A of our previous study [46]. For determining the tissue distribution of *acsl6* transcripts, tissue samples including eye, brain, gill, heart, liver, spleen, kidney, red/white muscle, mesenteric fat, anterior intestine, middle intestine, hind intestine, and ovary were collected from three fish. Tissue samples were frozen in liquid nitrogen immediately after collection and stored at −80 °C until analysis.

### 4.3. Cloning of acsl6 cDNA

Total RNA was extracted and mixed from common carp brain, liver, and muscle using Trizol reagent (Invitrogen, USA), and the first-strand cDNA was synthesised using Cloned AMV First-Strand cDNA Synthesis kit (Invitrogen, Carlsbad, CA, USA). To amplify the first fragment of *acsl6* cDNA, specific primers (*acsl6*-F1 and *acsl6*-R1, Table 1) were designed based on the common carp muscular transcriptome (unpublished data). PCR was performed using an RT-PCR kit from Invitrogen (Carlsbad, CA, USA) with an amplification program consisting of an initial denaturation at 94 °C for 3 min, followed by 30 cycles of denaturation at 94 °C for 30 s, annealing at 56 °C for 45 s and extension at 72 °C for 3 min, and a final extension at 72 °C for 10 min. The positive PCR fragments were confirmed by DNA sequencing (Shanghai Sangon Biotech Co., Ltd., China). Following this, the cDNA ends of the *acsl6* gene were obtained by 5′ and 3′ rapid amplification of cDNA ends using Gene Racer™ Kit (Invitrogen, Carlsbad, CA, USA) with gene-specific primers (*acsl6*-R2/*acsl6*-R3 and *acsl6*-F2/*acsl6*-F3, Table 2).

### 4.4. Sequence and Phylogenetic Analysis of acsl6

The deduced amino acid (aa) sequence of the newly cloned common carp *acsl6* cDNA was aligned with *acsl6* orthologues from other fish species, *Carassius auratus* (XP_026089889.1), *Danio rerio* (XP_021324602.1), *Epinephelus lanceolatus* (XP_033505573.1), *Nothobranchius furzeri* (XP_015816936.1), *Oncorhyncus mykiss* (XP_021479764.1), *Oreochromis niloticus* (XP_003451433.2), and *Salmo salar* (XP_014053481.1) by using ClustalW2. Multiple alignments of the corresponding *acsl6* aa sequence were performed using the EMBOSS Needle Pairwise Sequence Alignment tool (http://www.ebi.ac.uk/Tools/psa/emboss_needle/). The phylogenetic tree of *acsl6* from a variety of vertebrate lineages was constructed with the neighbor-joining method [47]. Confidence in the resulting phylogenetic tree branch topology was bootstrapped with 10,000 iterations.

### 4.5. Tissue Distribution of acsl6 Transcripts

The distributions of *acsl6* transcripts were measured in different tissues by reverse transcription PCR (RT-PCR) and real-time quantitative PCR (qPCR) analysis. Total RNAs from the brain, eyes, gills, heart, liver, kidney, spleen, red/white muscles, mesenteric adipose, anterior intestine, middle intestine, hind intestine, and ovary were extracted as described above, and 1 μg of RNA was reverse-transcribed into cDNA. RT-PCR was performed as described above. Additionally, qPCR was performed with a Lightcycler 480 system (Roche, Basel, Switzerland). The amplification was performed in 20 μL reaction system containing 5 μL of SYBR Premix, 0.5 μL of each primer (10 μM), 3 μL RNase free water, and 1 μL of cDNA. The real-time RT-PCR program was a two-step method: 1 cycle of 94 °C for 5 min; followed by 45 cycles at 95 °C for 10 s, annealing 60 °C for 20 s, extension 72 °C for 20 s; and with a final extension step at 95 °C for 5 s, 65 °C for 1 min, and 40 °C for 10 s. The expression of the housekeeping gene 18S rRNA was determined as an internal control. Triplicates of each reaction were performed for each sample. Primers used for RT-PCR and qPCR are shown in Table 1.

### 4.6. Cloning of acsl6 Gene Promoter and Construction of Deletion Mutants

Genomic DNA of muscle tissue was extracted from common carp using the standard phenol-chloroform extraction method and was used as a template for cloning the candidate promoter. The 1790-bp upstream sequence of the *acsl6* gene was obtained from the genomic sequencing data of common carp. For identifying the core promoter region of *acsl6* gene, four promoter fragments (SD1–SD4) were amplified using Transfer-PCR (TPCR) [48] with one of the specific forward primers (SD1-F, SD2-F, SD3-F, and SD4-F; Table 1) and a common reverse primer (SD-R; Table 1) and inserted into pGL3.0 vector (Promega, USA). The upstream sequence in the four insert fragments including SD4, SD3, SD2, and SD1 was of −1789 bp, −1231 bp, −758 bp, and −198 bp lengths to the putative transcription start site (TSS +1), respectively (Figure 4). These sequences were checked by sequencing in Shanghai Sangon Biotech Co., Ltd. (Shanghai, China).

### 4.7. Site-Directed Mutagenesis of acsl6 Core Promoter

The core promoter sequence of *acsl6* was analyzed by TRANSFAC^®^, MatInspector^®^, and JASPAR^®^. Bingding sites for eleven TFs including FOXO1, SP1, AP1, CDX1, YY1 (two), C/EBPα, TBP, SREBP1c, PPARα, and PPARγ were found in the core promoter (Figure 5). To detect the functions of these predicted TF binding sites on the activity of *acsl6* core promoter, site-directed mutation of recombinant plasmids (Table 3) was performed using Muta-directTM site-directed mutagenesis kit (SBS Genetech, Shenzhen, China) as described in the manufacturer’s protocol and were verified by DNA sequencing. The influence of TF binding site mutations on the promoter activity of *acsl6* was measured by dual-luciferase assay as described below, and the construct SD3 was used as a reference.

### 4.8. Cell Culture, Transfection, and Dual Luciferase Assay

HEK 293T cells were maintained in high glucose Dulbecco’s Modified Eagle Medium (Gluta-MAX DMEM) (Gibco, Life Technologies, New York, NY, USA) containing 10% fetal bovine serum (Gibco, New York, NY, USA). Cells were transfected using Lipofectamine^®^ LTX Reagent with PLUSTM reagent (Invitrogen, Carlsbad, CA, USA) following the manufacturer’s instruction. Briefly, the promoter reporter plasmid (100 ng) and the internal control vector pGL3.0 (0.01 ng) were co-transfected into the cells. The Dual Luciferase Reporter Assay System E2940 (Promega, Madison, WI, USA) was used to measure luciferase activity according to the manufacturer’s protocol. Briefly, cells were harvested 24 h after transfection and treated with 75 μL Dual-Glo Luciferase Assay Reagent (Promega, Madison, WI, USA) and incubated at 25 °C for 10 min. The promoter activity was calculated from the luminescence ratio of firefly/renilla luciferase for each construct. Each group was analyzed three times.

### 4.9. Electrophoretic Mobility Shift Assay (EMSA)

To investigate the interaction of TFs and the core promoter of the *acsl6* gene, the nuclear and cytoplasmic proteins of muscle were extracted with the Beyotime Nuclear Extract Kit (Beyotime, Shanghai, China), followed by quantification with Non-Interference Protein Assay Kit (Sangon, Shanghai, China). The 5′ end biotin-labeled probe (EMF1/EMR1, Table 1) of 364 bp covering TF elements (SP1, AP1, CDX1, YY1, C/EBPα, TBP, SREBP1, PPARα, and PPARγ) was designed and incubated with the muscular proteins. Both the labeled and unlabeled probes in the experiment were synthesized from Shanghai Sangon Biotech Co., Ltd. The EMSA reaction system was performed using the Beyotime Chemiluminescent EMSA Kit (Beyotime Institute of Biotechnology, Shanghai, China) as described in the manufacturer’s protocol. The detailed experimental condition of EMSA was according to our previous studies [10,49].

### 4.10. Intraperitoneal Injection Experiments

Two hundred forty healthy juvenile common carp were weighed and randomly distributed into four groups (twelve tanks with 20 fish each tank). After the acclimation, the fish were treated with dimethyl sulfoxide (DMSO, vehicle) (Sigma, St. Louis, MO, USA), PPARα agonist fenofibrate (Sigma, St. Louis, MO, USA), PPARγ agonist balaglitazone (Sigma, St. Louis, MO, USA), or SREBP1 antagonist betulin (Sigma, St. Louis, MO, USA). These agonists/antagonists were dissolved in 2.5% DMSO and diluted with 0.65% NaCl to obtain a final concentration of 1 mg/mL. All fish were fasted on the day before injection. Before injection, fish were first anesthetized with 0.01% 2-phenoxyethanol. The control group was treated with 2.5% DMSO, while the fenofibrate, balaglitazone, and betulin groups were treated with the corresponding agonists/antagonists (1 mg drug per 100 g fish). The drugs were slowly injected into their abdominal cavity. After the first injection, fish were put back into the corresponding tanks. After 24 h, each group was fed with the formula feed as mentioned above, and 6 days after the first injection, a similar dose of injection was repeated. At 24 h after the second injection, three fish from each tank (nine fish per group) were anesthetized with 0.01% 2-phenoxyethanol and muscle tissues were sampled from each fish, frozen in liquid nitrogen, and stored at −80 °C before the analyses of *acsl6*, *ppar*α, *ppar*γ, and *srebp*1 mRNA expression by qPCR (primers are shown in Table 1).

### 4.11. Analysis of Fatty Acids Composition

Total lipids were extracted from diets and muscle of common carp with a mixture of chloroform/methanol (*v*/*v*, 2:1), followed by fatty acid methyl esters (FAME) using boron trifluoride diethyl etherate (ca. 48%, Acros Organics, Waltham, MA, USA). FAME were analysed and separated using a gas chromatograph (GC2010-plus, Shimadzu, Japan) as previously described [50]. Briefly, FAME samples were applied using on-column injection, and the oven temperature was programmed from 80 to 250 °C at a rate of 40 °C min^−1^ and held at 210 °C for 30 min. Individual FAME were identified through comparison with commercial standards (Sigma, St. Louis, MO,) and quantified with CLASS-GC2010-plus workstation (Shimadzu, Kyoto, Japan).

### 4.12. Statistical Analysis

All data were expressed as means ± standard error of mean (SEM). One-way analysis of variance (ANOVA) followed by Tukey’s multiple comparison test was used to analyze the tissue distribution and promoter activity of *acsl6* gene. The effects of agonists/antagonists on *acsl6*, *ppar*α, *ppar*γ, and *srebp*1 mRNA expression were analyzed by the Student’s t-test. *p* values < 0.05 were considered statistically significant. All statistical analyses were performed using SPSS v17.0 (SPSS Inc., Michigan Avenue, Chicago, IL, USA).

## Figures and Tables

**Figure 1 ijms-21-04736-f001:**
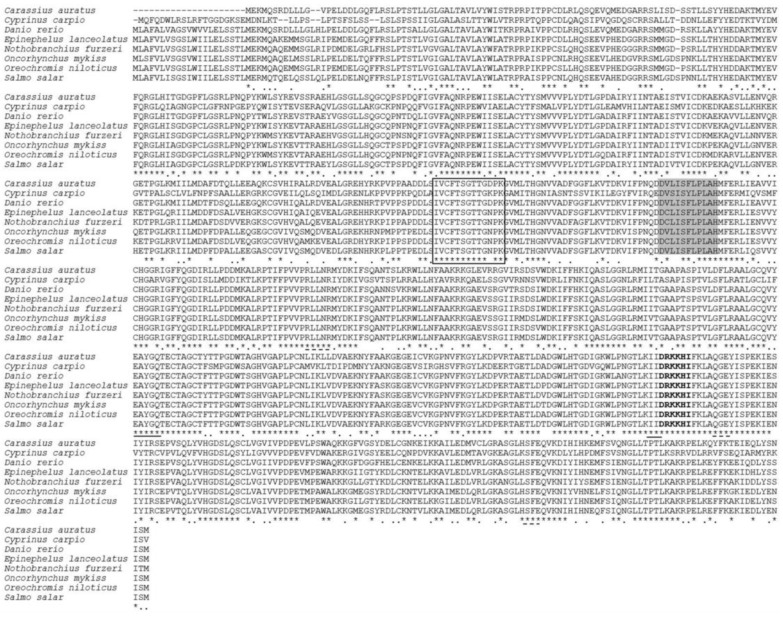
Alignment of the deduced amino acid (aa) sequences of *acsl6* isolated from *Cyprinus carpio* with other orthologues, including *Carassius auratus* (XP_026089889.1), *Danio rerio* (XP_021324602.1), *Epinephelus lanceolatus* (XP_033505573.1), *Nothobranchius furzeri* (XP_015816936.1), *Oncorhyncus mykiss* (XP_021479764.1), *Oreochromis niloticus* (XP_003451433.2), and *Salmo salar* (XP_014053481.1): Deduced aa sequences were aligned using ClustalW2. Identical and similar residues are marked with * and:, respectively. The conserved acyl-activating enzyme (AAE) motif is a black box, the gate domain is shaded, two putative CoA binding sites are dash-underlined, the two putative AMP binding sites are solid underlined, and the linker domain is represented with bold.

**Figure 2 ijms-21-04736-f002:**
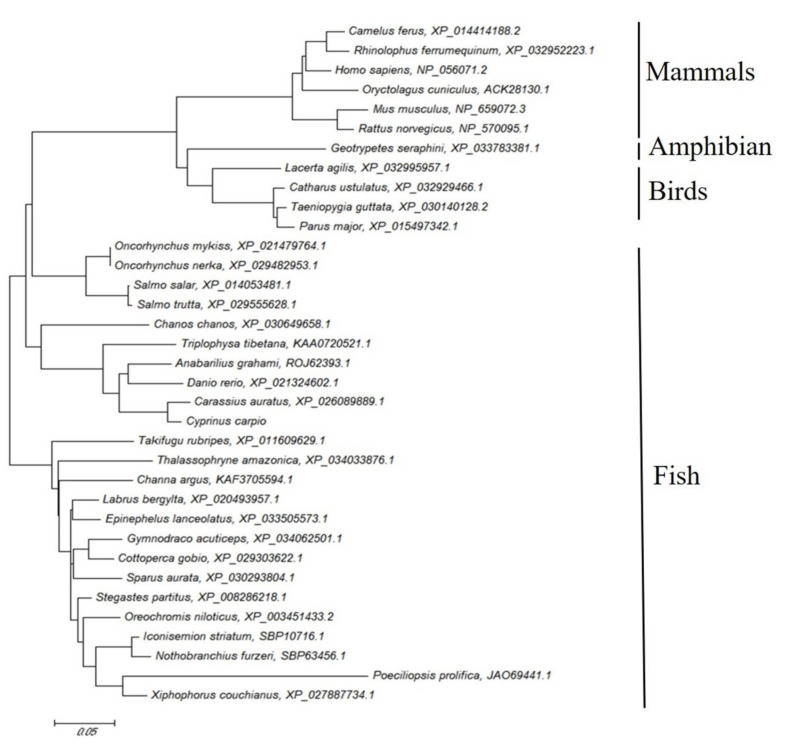
Phylogenetic analysis of *acsl6* from different animal species: The tree is based on an alignment corresponding to full-length amino acid sequences using ClustalW and MEGA (5.0). The number denote the bootstrap majority consensus values from 1000 replicates.

**Figure 3 ijms-21-04736-f003:**
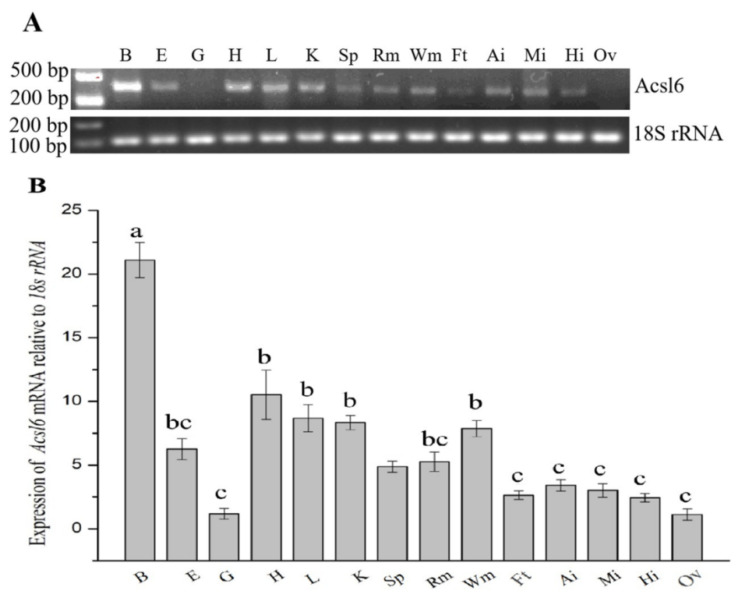
Semi-quantitative RT-PCR (**A**) and real-time quantitative PCR (**B**) analyses of *acsl6* mRNA expression: Expression values were normalized to those of 18S rRNA. Data are means ± SEM (*n* = 3). Bars not sharing a common letter indicated significant differences (*p* < 0.05) as determined by one-way ANOVA followed by Tukey’s multiple comparison test. Abbreviations: B, brain; E, eye; G, gill; H, heart; L, liver; K, kidney; Sp, spleen; Rm, red muscle; Wm, white muscle; Ft, mesenteric fat; Ai, anterior intestine; Mi, middle intestine; Hi, hind intestine; and Ov, ovary.

**Figure 4 ijms-21-04736-f004:**
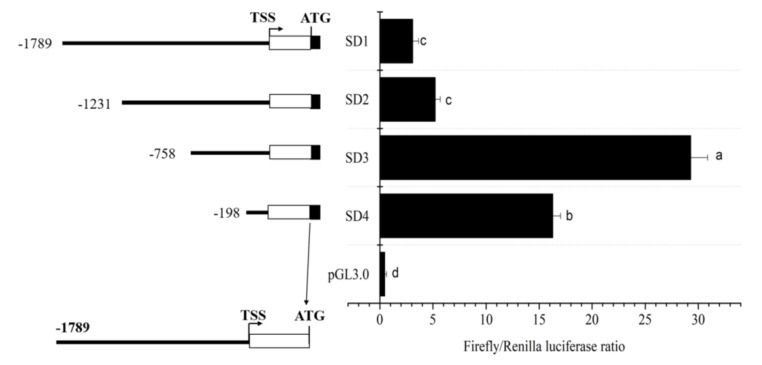
Structure and progressive deletion analysis of grouper *acsl6* promoter: 5′ deletion constructs are shown on the upper left, and the structure of *acsl6* promoter is showed on the lower left. Noncoding exons are indicated with open boxes, and luciferase coding frames are indicated by closed boxes. The sequence is numbered relative to the first base of the transcription start site (TSS), assumed to be the first base of the 5′ noncoding exon. Promoter activity of constructs is represented on the right with the values representing normalized activity (Firefly luciferase–Renilla luciferase). Bars not sharing a common letter indicated significant differences (*p* < 0.05) among deletions determined by one-way ANOVA followed by Tukey’s multiple comparison test.

**Figure 5 ijms-21-04736-f005:**
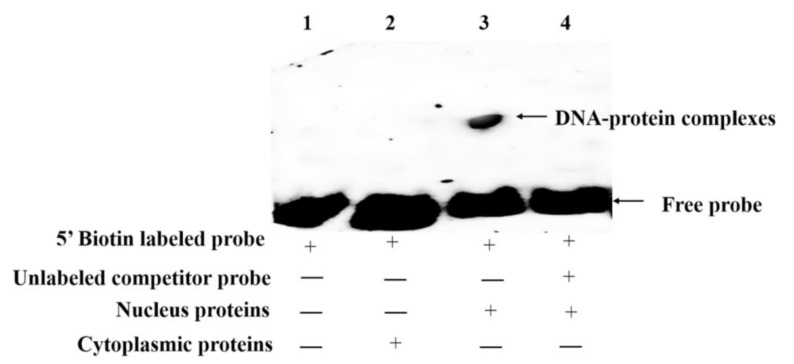
Electrophoretic mobility shift assay (EMSA) of the common carp *acsl6* core promoter with muscular nuclear proteins: Each lane is represented as lane 1 (no proteins, 5′ biotin labeled probe), lane 2 (muscular cytoplasmic protein and 5′ biotin labeled free probe), lane 3 (muscular nucleoprotein and 5′ biotin labeled free probe), and lane 4 (muscular nucleoprotein, unlabeled competitor probe, and 5′ biotin labeled free probe). “+” means that the corresponding material in the row has been added, and “−” means that the material is not added.

**Figure 6 ijms-21-04736-f006:**
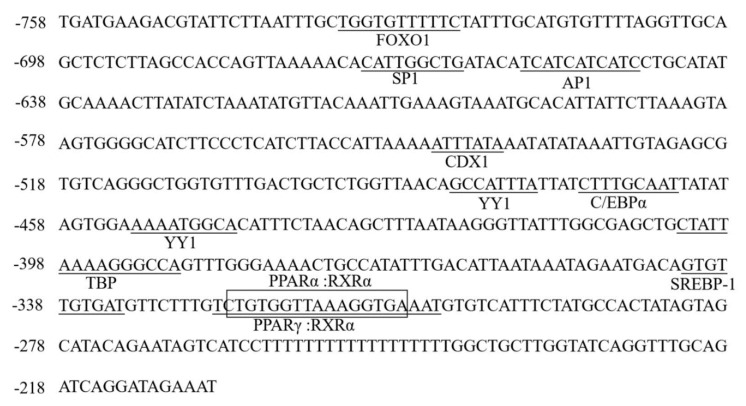
The nucleotide sequence and predicted binding sites for transcription factors in the core region of common carp *acsl6* promoter: Numbers are given relative to the first base of the transcription start site (TSS). Potential transcription binding motifs are marked in underline and open boxes for PPARα: RXRα. Details for the name of transcription factors are shown in the text.

**Figure 7 ijms-21-04736-f007:**
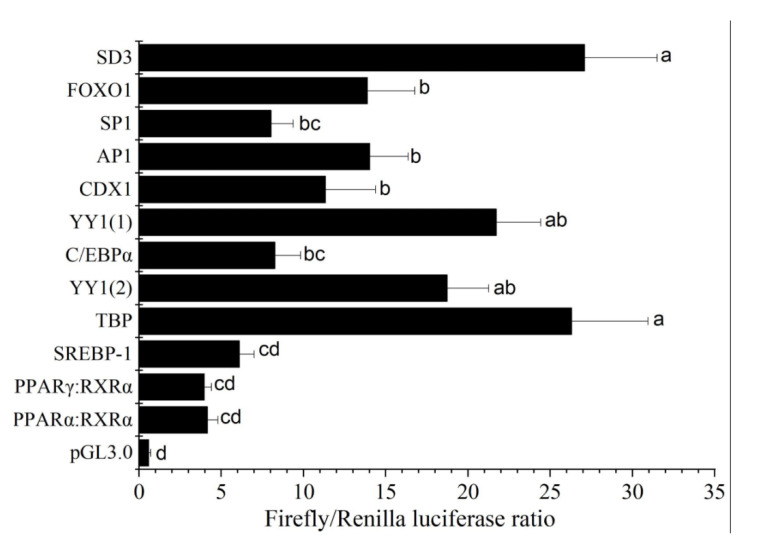
The effects of site-directed mutation of transcription factor binding sites on the promoter activity of common carp *acsl6* gene: Mutations of promoter deletion on SD3 (wild type, −758 to ATG) were generated according to in silico predications as shown in Figure 4, and the effects of mutation on promoter activity were compared with SD3. Promoter activity of constructs is represented on the right with the values representing normalized activity (Firefly luciferase–Renilla luciferase). Bars not sharing a common letter indicated significant differences (*p* < 0.05) among deletions determined by one-way ANOVA followed by Tukey’s multiple comparison test.

**Figure 8 ijms-21-04736-f008:**
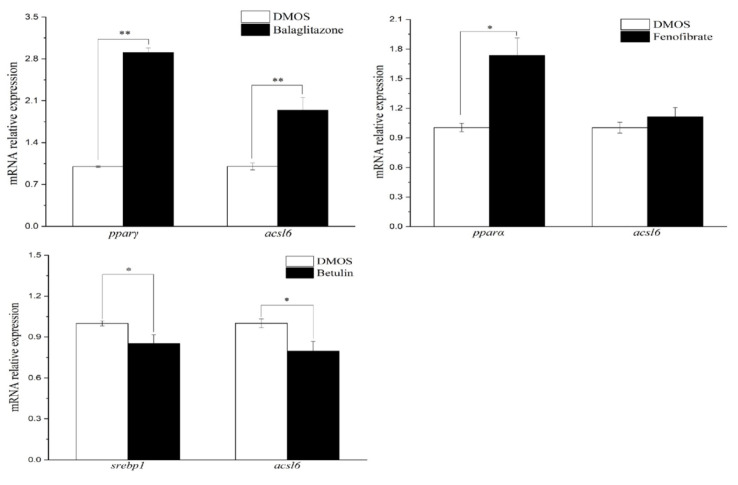
Q-PCR analysis of *acsl6*, *pparα*, *pparγ*, and *srebp1* mRNA expression levels in the muscle of juvenile common carp injected with Ppar*α* agonist (fenofibrate), Pparγ agonist (balaglitazone), Srebp1 antagonist (betulin), or control (dimethyl sulfoxide (DMSO)). Results are means ± SEM (*n* = 3). Significant differences compared between the injected treatments (fenofibrate, balaglitazone, and betulin), and corresponding control group were analyzed using Student’s t-test, with * denoting *p* < 0.05 and with ** denoting *p* < 0.01.

**Table 1 ijms-21-04736-t001:** Main fatty acids composition in muscle of common carp injected with siRNA, agonists (balaglitazone and fenofibrate), and antagonist (betulin) (% area).

Main Fatty Acids	Injected Treatments			
	DMSO	Fenofibrate	Balaglitazone	Betulin
16:0	13.10 ± 0.68	11.92 ± 1.78	12.22 ± 2.32	14.73 ± 1.51
18:0	1.54 ± 0.05	1.11 ± 0.07 *	1.05 ± 0.05 *	1.93 ± 0.07 *
20:0	0.28 ± 0.03	0.21 ± 0.02	0.25 ± 0.02	0.23 ± 0.01
16:1	1.99 ± 0.18	2.10 ± 0.37	2.13 ± 0.17	2.31 ± 0.29
18:1	25.13 ± 1.44	21.81 ± 3.11	23.25 ± 3.15	27.49 ± 4.06
18:2n-6	32.92 ± 3.54	33.92 ± 3.51	31.97 ± 2.13	34.11 ± 2.34
18:3n-6	0.23 ± 0.05	0.28 ± 0.10	0.27 ± 0.04	0.13 ± 0.01 *
20:4n-6 (ARA)	1.54 ± 0.03	1.55 ± 0.22	1.63 ± 0.06	1.38 ± 0.10 *
18:3n-3	5.21 ± 0.02	4.93 ± 0.02	4.85 ± 0.01	5.16 ± 0.02
18:4n-3	0.43 ± 0.10	0.40 ± 0.04	0.44 ± 0.03	0.35 ± 0.01
20:5n-3 (EPA)	0.43 ± 0.03	0.41 ± 0.02	0.48 ± 0.03	0.39 ± 0.01
22:6n-3 (DHA)	2.27 ± 0.10	2.28 ± 0.13	2.53 ± 0.07 *	1.81 ± 0.11 *

Results are means ± SEM (*n* = 3). Significant differences compared between the injected treatments (fenofibrate, balaglitazone, and betulin) and corresponding control group were analyzed using Student’s *t*-test, with * denoting *p* < 0.05.

**Table 2 ijms-21-04736-t002:** PCR primers sequence and RNAi nucleotide sequence used in this study.

Subject	Primer Name	Primer Sequence (5′–3′)
First fragment cloning	*acsl6*-F1	ATGCAGTTTCAGGATTGGTTACGT
	*acsl6*-R1	CTATACGGATGATTTCCTGTACATC
3′RACE	*acsl6*-F2	CAGGTGAAGGACTTATATTTGCAC
	*acsl6*-F3	GATGTACAGGAAATCATCCGTATAG
5′RACE	*acsl6*-R2	GCGGAGCGTCTACAGCTCTGATC
	*acsl6*-R3	ACGTAACCAATCCTGAAACTGCAT
PCR for 5′ flanking sequence cloning	SD1-F	CTTACGCGTGCTAGCCCGGGCTCGAGGCCCAGAATTTGCACAGATTACGTG
	SD2-F	GCTCTTACGCGTGCTAGCCCGGGCTCGAGTGTGAAACATTATGTAAGTTGAATACTTTTAC
	SD3-F	GCTCTTACGCGTGCTAGCCCGGGCTCGAGTGATGAAGACGTATTCTTAATTTGCTGG
	SD4-F	GCTCTTACGCGTGCTAGCCCGGGCTCGAGACTTATAGTTTGAATCAGATTTTATTTTCCAT
	SD-R	ACCAACAGTACCGGAATGCCAAGCTTCTTGCCATCTCCTCCAGTGAAAC
EMSA probes	EMF1 (5′-biotin labeled)	CATTGGCTGATACATCATCATCAT
	EMR1 (5′-biotin labeled)	CATTTCACCTTTAACCACAGACA
	EMF2	ATTGGCTGATACATCATCATCAT
	EMR2	ATTTCACCTTTAACCACAGACA
RT-PCR and qPCR	RT-*acsl6*-F	ATGCAGTTTCAGGATTGGTTACGT
	RT-*acsl6*-R	TGCTATCTGCAGTCCTCTCTGGAA
	*Q-acsl6*-F	GATCAGAGCTGTAGACGCTCCGCT
	*Q-acsl6*-R	CCTCAGTGTAAGAGATCCACTGAT
	*pparγ*-F	AGACATGGTGGACACGCAGACGTT
	*pparγ*-R	CGCTCTCCTGCATCCTGTAGTTCT
	*pparα*-F	GGGAAAGAGCAGCACGAG
	*pparα*-R	GCGTGCTTTGGCTTTGTT
	*srebp*1-F	CGTCTGCTTCACTTCACTACTC
	*srebp*1-R	GGACCAGTCTTCATCCACAAA
	18S rRNA	CTGAGAAACGGCTACCATTC
	18S rRNA	GCCTCGAAAGAGACCTGTATTG

**Table 3 ijms-21-04736-t003:** Primers used for site-directed mutations of transcription factors (TFs) on common carp *acsl6* core promoter.

TFs	Position	Predicted Element	Mutation Site
FOXO1	−733	TGGTGTTTTTC	TGGTGTTTTTC→×
SP1	−672	CATTGGCTG	CATTGGCTG→×
AP1	−658	TCATCATCATC	TCATCATCATC→×
CDX1	−545	ATTTATA	ATTTATA→×
YY1	−485	GCCATTTA	GCCATTTA→×
	−453	AAAAATGGCACA	AAAAATGGCACA→×
C/EBPα	−474	CTTTGCAAT	CTTTGCAAT→×
TBP	−404	CTATTAAAAGGGCCA	CTATTAAAAGGGCCA→×
SREBP-1	−343	GTGTTGTGATG	GTGTTGTGATG→×
PPARγ:RXRα	−325	TCTGTGGTTAAAGGTGAAAT	TCTGTGGTTAAAGGTGAAAT→×
PPARα:RXRα	−323	TGTGGTTAAAGGTGAA	TGTGGTTAAAGGTGAA→×

Notes: The binding sites for TFs are described in Figure 5. The bases underlined are chosen for site-directed mutant (deletion), “→” means substitution, and “×” means deletion.

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
