# Peer review of "Molecular Cloning, Characterization, and Expression Regulation of Acyl-CoA Synthetase 6 Gene and Promoter in Common Carp Cyprinus carpio"

_ijms, 2020, doi:10.3390/ijms21134736_

Round 1

Reviewer 1 Report

Xie et al present a well written and thorough molecular analysis to demonstrate the existence of Acsl6 in fish, its expression across tissues, and its molecular regulation by transcription factors. The data are well presented and straightforward. The impact of the work is directly related to the major source of dietary DHA, i.e. fish, and how fish regulates DHA content. There were several concerns presented as outlined below:

Major

  • The authors mention that this work is relevant to the difference in fish DHA content between farmed vs wild caught. However, it is not clear how the work presented relates to this issue directly. Is the presumption that changes in Acsl6 regulation are causing the difference between farmed and wild fish DHA content, or is it supply, or a combination? Perhaps comparison of wild to farmed in acsl6 content in relation to DHA content is appropriate.It is better to address this problem either experimentally or in the discussion.
  • The authors should include biochemical/enzyme activity description of the ACSs in the introduction.

Minor

  • Several minor typos throughout paper, see lines: 15-16 (tense), 49 (awkward), 156 (extra spacing), 234 (too many “expression”), 267 (change “the”)
  • References are numbered twice.

Author Response

Many thanks for your comments, which is very helpful and has enabled us to revise and improve the manuscript (ijms-849757). The point-to-point responses your comments are provided below.

  1. The authors mention that this work is relevant to the difference in fish DHA content between farmed vs wild caught. However, it is not clear how the work presented relates to this issue directly. Is the presumption that changes in Acsl6 regulation are causing the difference between farmed and wild fish DHA content, or is it supply, or a combination? Perhaps comparison of wild to farmed in acsl6 content in relation to DHA content is appropriate.It is better to address this problem either experimentally or in the discussion.

Response: Thanks for your suggestion. To date, the plant lipid sources (without DHA) are widely applied in common carp farming industry to completely replace the fish oil. Thus, farmed common carp lose the chance of acquiring the dietary DHA compared with wild fish, which is the important reason of low DHA level in farmed fish.  

Indeed, there is no denying that dietary fatty acid compositions also cause the difference of Acsl6 regulation between farmed and wild fish, which may further reduce the DHA content in farmed fish. In future work, we will study the effects of dietary fatty acid compositions on the regulation of Acsl6. Accordingly, these statements have been inserted in the discussion section. Details please see the Revised manuscript.

  1. The authors should include biochemical/enzyme activity description of the ACSs in the introduction.

Response: ACSL enzymes play an important role in lipid metabolism since these enzymes activate long chain fatty acid (LCFA, with 12-22 carbons) to form acyl-CoAs, which is subsequently used in almost all catabolic and anabolic downstream pathways of FA metabolism. Accordingly, these statements have been inserted in the introduction, and the appropriate reference has been added to the References section.

  1. Several minor typos throughout paper, see lines: 15-16 (tense), 49 (awkward), 156 (extra spacing), 234 (too many “expression”), 267 (change “the”)

Response: According to your suggestion, the corrections have done in the revised MS. 

  1. References are numbered twice.

Response: The twice numbered references are probably because the manuscript has been edited and formatted by editors. The second number is the real number of references, but the first number cannot be deleted by us.

Reviewer 2 Report

The experiments are well designed and performed. The presentation of the data is clear. Results are convincing and publication is recommended. I have some minor comments to improve the manuscript before to be published:

Line 60-61: Revise the sentence

Line 70: significantly lower than

Line 286-287: Revise the sentence

Author Response

Many thanks for your comments, which is very helpful and has enabled us to revise and improve the manuscript (ijms-849757). The point-to-point responses to your comments are provided below.

  1. Line 60-61: Revise the sentence

Response: Thanks for your suggestion, “Additionally, the variety of ACSLs in mammals is associated with distinct substrate preferences.” was revised as “Additionally, five distinct ACSL genes differing in their substrate preferences have been identified in mammals” in the L60-61.

  1. Line 70: significantly lower than

Response: According to your suggestion, “significantly low than” was revised as “significantly lower than”.

  1. Line 286-287: Revise the sentence

Response: “All surgeries were performed under anesthesia using 0.01% 2-phenoxyethanol (Sigma-Aldrich, St. Louis, MO, USA)” was reworded into “All the fish surgeries were performed with 0.01% 2-phenoxyethanol (Sigma-Aldrich, St. Louis, MO, USA) anesthesia to minimize suffering”